# A tranquil virtual reality experience to reduce subjective stress among COVID-19 frontline healthcare workers

Elizabeth Beverly[1,2]*, Laurie Hommema[1,3], Kara Coates[4], Gary Duncan[3,4,5], Brad Gable[1,4,5], Thomas Gutman[4,5], Matthew Love[6,7], Carrie Love[7,8], Michelle Pershing[9], Nancy Stevens[1,10]

1 Ohio University Heritage College of Osteopathic Medicine, Athens, Ohio, United States of America, 2 Ohio University Diabetes Institute, Athens, Ohio, United States of America, 3 OhioHealth Riverside Methodist Hospital, Columbus, Ohio, United States of America, 4 OhioHealth Healthcare Organization, Columbus, Ohio, United States of America, 5 OhioHealth Center for Medical Education & Innovation, Columbus, Ohio, United States of America, 6 Ohio University J. Warren McClure School of Emerging Communication Technologies, Athens, Ohio, United States of America, 7 Ohio University Game Research and Immersive Design (GRID) Lab, Athens, Ohio, United States of America, 8 Hocking College, Nelsonville, Ohio, United States of America, 9 OhioHealth Research Institute, Columbus, Ohio, United States of America, 10 Ohio University Museum Complex, Athens, Ohio, United States of America

☯ These authors contributed equally to this work.
* beverle1@ohio.edu

**Data Availability Statement:** The data underlying the results presented in the study are available from the OhioHealth Research Institute (contact: 1.614.566.1250; https://www.ohiohealth.com/

## Abstract

### Objective

The novel coronavirus-19 (COVID-19) has taken an immense physical, social, and emotional toll on frontline healthcare workers. Research has documented higher levels of anxiety, depression, and burnout among healthcare workers during the pandemic. Thus, creative interventions are needed now more than ever to provide brief, accessible support to frontline workers. Virtual reality is a rapidly growing technology with potential psychological applications. In this study, we piloted a three-minute Tranquil Cinematic-VR simulation of a nature scene to lower subjective stress among frontline healthcare workers in COVID-19 treatment units. We chose to film a nature scene because of the extensive empirical literature documenting the benefits of nature exposure and health.

### Methods

A convenience sample of frontline healthcare workers, including direct care providers, indirect care providers, and support or administrative services, were recruited from three COVID-19 units located in the United States. Inclusion criteria for participation included adults aged 18 years and older who could read and speak in English and were currently employed by the healthcare system. Participants viewed a 360-degree video capture of a lush, green nature preserve in an Oculus Go or Pico G2 4K head-mounted display. Prior to viewing the simulation, participants completed a brief demographic questionnaire and the visual analogue scale to rate their subjective stress on a 10-point scale, with 1 = 'Not at all stressed' to 10 = 'Extremely stressed.' We conducted paired t-tests to examine pre- and

ohiohealth-research-and-innovation-institute/
about-us).

**Funding:** The authors received no specific funding
for this work.

**Competing interests:** The authors have declared
that no competing interests exist.

post-simulation changes in subjective stress as well as Kruskal-Wallis tests and Mann-Whit-
ney U tests to examine differences by demographic variables. All analyses were conducted
in SPSS statistical software version 28.0. We defined statistical significance as a p-value
less than .05.

## Results

A total of 102 individuals consented to participate in the study. Eighty-four (82.4%) partici-
pants reported providing direct patient care, 73 (71.6%) identified as women, 49 (48.0%)
were between the ages of 25–34 years old, and 35 (34.3%) had prior experience with VR.
The pre-simulation mean stress score was 5.5±2.2, with a range of 1 to 10. Thirty-three
(32.4%) participants met the 6.8 cutoff for high stress pre-simulation. Pre-simulation stress
scores did not differ by any demographic variables. Post-simulation, we observed a signifi-
cant reduction in subjective stress scores from pre- to post-simulation (mean change = -2.2
±1.7, t = 12.749, p < .001), with a Cohen's d of 1.08, indicating a very large effect. Further,
only four (3.9%) participants met the cutoff for high stress after the simulation. Post-simula-
tions scores did not differ by provider type, age range, gender, or prior experience with vir-
tual reality.

## Conclusions

Findings from this pilot study suggest that the application of this Tranquil Cinematic-VR
simulation was effective in reducing subjective stress among frontline healthcare work-
ers in the short-term. More research is needed to compare the Tranquil Cinematic-VR
simulation to a control condition and assess subjective and objective measures of stress
over time.

## Introduction

On March 11, 2020, the World Health Organization (WHO) declared the novel coronavirus-
19 (COVID-19) a global pandemic [1]. Since that time, healthcare workers have risked their
lives fighting on the frontlines in the battle against COVID-19. Healthcare workers, particu-
larly those working in critical care, emergency medicine, infectious disease, and pulmonary
medicine, have put themselves at greater risk for infection, serious illness, and even death [2–
7]. Combined with increased workloads, shortages in personal protective equipment (PPE),
fears of infecting families, friends, and colleagues, and financial strains on healthcare organiza-
tions, COVID-19 has taken an immense physical, social, and emotional toll on healthcare
workers. Research has documented higher levels of anxiety, depression, and burnout among
healthcare workers during the pandemic [8–12]. Moreover, preliminary research suggests
healthcare providers may be at increased risk for post-traumatic stress disorder [13]. These
findings are concerning given professional burnout and mental health issues among healthcare
workers were already at epidemic levels prior to the pandemic [14–17]. For these reasons, crea-
tive solutions that provide psychological support to frontline healthcare workers are needed
now more than ever.

Virtual reality (VR) is a rapidly growing technology with potential psychological applica-
tions to support healthcare workers. VR is a set of technologies, including head-mounted

displays (HMDs), computers, and mobile devices, that simulate real-world objects, events, locations, and interactions in three-dimensional (3D) sensory environments [18, 19]. VR is classified as immersive or non-immersive [20]. Immersive VR transports the user from the physical world to a virtual environment by occluding visual contact with the external word; in contrast, non-immersive VR does not fully occlude the external world [20]. Two recent meta-reviews by Riva et al. in 2016 [21] and 2019 [22] demonstrated the clinical potential of VR in the treatment of several mental health disorders, including generalized anxiety disorder [23], panic disorder [24], social anxiety disorder [24], post-traumatic stress disorder [24], addiction [25], eating disorders [26, 27], depression [28], and stress management [29]. Although these findings are promising, the duration (1–16 sessions), length (3–60 minutes), and location (research laboratory, clinic, or hospital) of these VR interventions may be a significant obstacle to widespread implementation. Thus, to meet the needs of frontline healthcare workers new VR interventions must be brief and easily accessible.

For the present study, we created a three-minute Tranquil Cinematic-VR (Cine-VR) simulation of a nature scene to lower stress among frontline healthcare workers in COVID-19 treatment units. We chose to film a nature scene because of the extensive empirical literature documenting the benefits of nature exposure and health [30, 31]. The purpose of our study was to pilot the effectiveness of the Tranquil Cine-VR simulation in lowering subjective stress scores. We hypothesized that the Tranquil Cine-VR simulation would lower subjective stress scores among the frontline healthcare workers.

## Materials and methods

The purpose of this pilot study was to create a Tranquil Cine-VR simulation to reduce subjective stress among frontline workers employed at COVID-19 treatment units located in the midwestern United States. Cine-VR leverages 360-video with the techniques of cinema to create VR [32]. Participants volunteered to view an immersive three-minute 360-degree Cine-VR scene of a local nature preserve via an Oculus Go or Pico G2 4K HMDs. We administered a brief visual analogue scale to assess changes in subjective stress before and after the Tranquil Cine-VR. The OhioHealth Office of Human Subjects Protections (Institutional Review Board #1734245–1) and the Ohio University Office of Research Compliance (Institutional Review Board #21-D-28) approved the protocol, recruitment procedures, and materials.

### Recruitment

A convenience sample of frontline healthcare workers, including direct care providers, indirect care providers, and support or administrative services, were recruited from three distinct locations of the OhioHealth Healthcare System in the United States during the spring of 2021. The three COVID-19 units included an emergency department, a medical/surgical unit, and a critical care unit. Participants were recruited via well-being partners located at each site in the staff rooms. OhioHealth hired well-being partners to promote health and resilience for the frontline workers during the COVID-19 pandemic. All participants provided verbal informed consent to participate in the assessment. Inclusion criteria for participation included adults aged 18 years and older who could read and speak in English and were currently employed by the healthcare system. There were no other exclusion criteria.

### Power analysis

We conducted an a priori power analysis using Statulator [33], an online statistical calculator, which determined a total sample size of 90 participants was estimated for 80% power at a 5% significance level (two-sided, $P < .05$) to detect an effect size of 0.3 in the paired data.

### Cinematic 360-degree virtual reality simulations (cine-VR)

Technological advancements in 360-video capture offer filmmakers an opportunity to leverage the techniques of cinema to 360-video. In cine-VR, participants view a film within a VR HMD to experience a world in a novel way. This melding of cinema and VR gave rise to the new term, Cine-VR. Cine-VR is an immersive, realistic new way to experience nature by connecting viewers in ways never before possible.

The Tranquil Cine-VR simulation represented a collaboration among the Ohio University's Game Research and Immersive Design (GRID) Lab, the OHIO Museum Complex, the Ohio University Heritage College of Osteopathic Medicine, and the OhioHealth Healthcare System. The interdisciplinary team consisted of filmmakers, VR simulation experts, ecologists, environmentalists, health behaviorists, researchers, and physicians. The Cine-VR simulation depicted a lush, green nature preserve to promote relaxation and peace. For the purposes of this study, the Cine-VR simulation was three minutes in length. The brevity of the simulation was designed to provide frontline workers with a brief respite during their work breaks. The simulation was screened in an Oculus Go or Pico G2 4K HMD so that participants could turn their head and body in any direction and gather relevant information, much as if they were present in the actual location. Observant participants could notice subtle details, such as the way the sun filtered through the trees, the different plants growing throughout the forest, or the moss growing on rocks co-occurring in the space. Further, the participants could hear the sounds of birds chirping and leaves rustling in the wind. With traditionally shot films, this information would be presented in a close-up or with camera movement to call a participant's attention to relevant information, resulting in a more passive and guided viewing experience. Presenting the content in cine-VR creates an active viewing experience with participants choosing what they want to look at and pay attention to, increasing immersion and encouraging emotional engagement. Participants may feel a sense of accomplishment as they notice subtle details planted by the filmmaking team, heightening the experience of being present in the virtual world. The HMDs were sanitized after each use with Clean-box ultraviolet light surface decontamination boxes (Cleanbox, Nashville, TN).

### Measures

Participants completed a series of demographic questions that assessed age range, gender, level of patient care, and prior experience with VR. The participants' exact age, race, ethnicity, and provider type were not collected because they were deemed identifiable information per the Office of Human Subjects Protections. Participants also completed the visual analogue scale (VAS) to rate their subjective stress on a 10-point scale, with 1 = 'Not at all stressed' to 10 = 'Extremely stressed' [34, 35]. The VAS shows a high level of correlation with the well-validated Perceived Stress Scale (PSS)-14 [35]. In addition, a VAS threshold of 6.8 has the discriminant power to predict a high stress level via a PSS-14 cutoff score of $\geq 7.2$ [35, 36]. The demographic questions and VAS were administered using REDCap, a secure, electronic data capture program designed for research hosted by OhioHealth Research Institute [37, 38].

### Data collection

All participants provided verbal informed consent prior to participation in the assessment. We selected verbal consent because the study was conducted in three COVID-19 units, and we wanted to limit the use of materials to prevent the spread of the virus. Both the OhioHealth Office of Human Subjects Protections (Institutional Review Board #1734245–1) and the Ohio University Office of Research Compliance (Institutional Review Board #21-D-28) approved the verbal consent process.

Participants either read the verbal informed consent form on an iPad or had the verbal informed consent form read to them by a well-being partner. The informed consent form emphasized the voluntary nature of participation and reminded individuals that the study was not related to their employment. Importantly, workers did not have to participate in the study to view the Tranquil Cine-VR simulation. Workers who provided their verbal consent to the well-being partner were provided an iPad to complete the anonymous, electronic REDCap survey. Participants completed the REDCap survey before and after viewing the Tranquil Cine-VR simulation. The estimated amount of time to finish the REDCap survey questions was one to two minutes. The data collection was intended to be brief for participants to complete during a break with minimal survey burden. The iPads were sanitized after each use with medical disinfectant wipes by the participants or well-being partners.

## Statistical analysis

Initial analyses examined the distribution and other descriptive data to determine that data met the assumptions of statistical tests. Next, we assessed demographic factors using descriptive statistics and presented them as sample sizes and percentages. We conducted nonparametric tests, Kruskal-Wallis tests and Mann-Whitney U tests, to examine differences by provider type, age range, gender, and prior VR experience. We performed paired t-tests to examine changes in perceived stress before and after the Tranquil Cine-VR simulation. In addition, we determined effect sizes using Cohen's d by calculating the mean difference between the pre- and post-assessment responses divided by the pooled standard deviation. We defined statistical significance as a p-value less than .05 and conducted analyses in SPSS statistical software version 28.0 (Chicago, IL: SPSS Inc.).

## Results

A total of 102 individuals consented to participate in the study. Eighty-four (82.4%) participants reported providing direct patient care, seven (6.9%) reported indirect patient care, and 10 (9.8%) reported support or administrative services. Seventy-three (71.6%) participants identified as women, 49 (48.0%) were between the ages of 25–34 years old, and 35 (34.3%) had prior experience with VR (Table 1).

### Pre-simulation subjective stress

The pre-simulation mean stress score was 5.5±2.2, with a range of 1 to 10. Thirty-three (32.4%) participants met the 6.8 cutoff for high stress pre-simulation (Fig 1). Pre-simulation stress scores did not differ by provider type (H(2) = 2.762, p = .251), age range (H(4) = 3.930, p = .416), gender (F(2) = 5.798, p = .055), or prior experience with VR (U = 1166.5, p = .966).

### Post-simulation subjective stress

Post-simulation, the mean stress score was 3.3±1.8, with a range of 1 to 10. We observed a significant reduction in subjective stress scores from pre- to post-simulation (mean change = -2.2 ±1.7, t = 12.749, p < .001), with a Cohen's d of 1.08, indicating a very large effect. Only four (3.9%) participants met the cutoff for high stress after the simulation, and the number of people with high stress differed pre- and post-simulation ($\chi^2$ = 8.582, p = .003). Additionally, participants who met the cutoff for high stress pre-simulation showed a greater reduction in subjective stress scores compared to participants who did not meet the cutoff pre-simulation (mean change = 3.3±2.0 vs. 1.6±1.2, t = 5.403, p < .001; Fig 1). Post-simulation stress scores and mean change scores, respectively, did not differ by provider type (H(2) = .050, p = .975; H

**Table 1. Participant demographic characteristics (n = 102).**

| Variable | n (%) |
|---|---|
| Age | |
| 18–24 years | 12 (11.8) |
| 25–34 years | 49 (48.0) |
| 35–44 years | 20 (19.6) |
| 45–54 years | 15 (14.7) |
| 55 to 64 years | 6 (5.9) |
| 65 years and older | 0 (0) |
| Gender | |
| Woman | 73 (71.6) |
| Man | 28 (27.5) |
| Other | 1 (1.0) |
| Level of patient care | |
| Direct patient care | 84 (82.4) |
| Indirect patient care | 7 (6.9) |
| Support or administrative services | 10 (9.8) |
| Prior experience with virtual reality | |
| Yes | 35 (34.3) |
| No | 67 (65.7) |

(2) = 3.460, p = .177), age range (H(4) = 8.951, p = .062; H(4) = 1.306, p = .860), gender (H(2) = 2.291, p = .318; H(2) = 3.513, p = .173), or prior experience with VR (U = 1106.0, p = .809; U = 1130.0, p = .947; Fig 2).

## Discussion

Experiences in nature and green environments have been shown to have a positive impact on human health and have even been suggested as a way to address health provider burnout [39]. Often set within the context of the Japanese practice of *shinrin yoku*, or forest bathing, numerous studies have explored how nature immersion can enable people of all ages to reduce stress

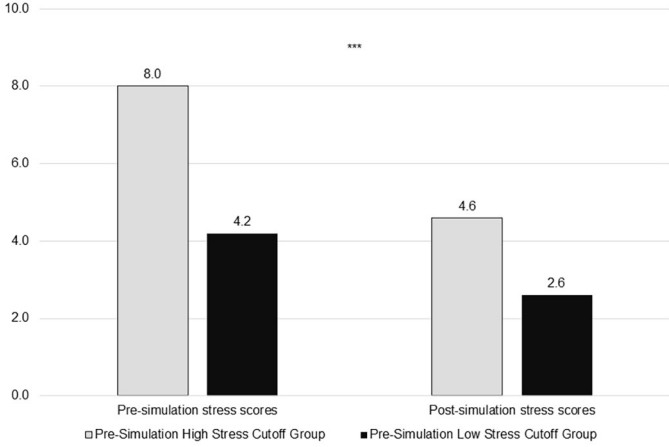

**Fig 1. Pre- and post-simulation subjective stress scores between high and low stress cutoff groups (n = 101).** ***p < .001.

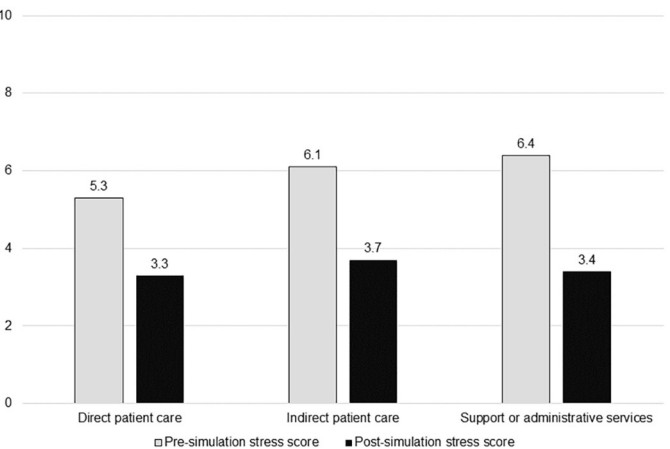

**Fig 2. Pre- and post-simulation subjective stress scores by provider type, age range, gender, and prior experience with virtual reality (n = 101).**

and enhance health and wellbeing [39, 40]. A number of positive physiological and psychological effects of mindfulness and nature immersion have been documented, with benefits including decreased heart rate [40], lower systolic and diastolic blood pressure [41, 42], reduced salivary [43] and serum [44] cortisol levels, lower urinary adrenaline [44], higher serum adiponectin [45], and stronger immune responses [46, 47]. Notably, nature immersion has been associated with significantly lower Profile of Mood State (POMS) [42, 44, 45] and Total Mood Disorder (TMD) scores [44]. Numerous studies self-reported relief from depression and/or an increased sense of well-being [48–51] resulting in marked positive mental health impacts in high-risk (highly stressed) individuals [50]. Although most of these studies have focused on immersive experiences in natural settings combined with physical activity [50] and/or mindfulness prompts [43], a study exploring physiological responses to viewing natural landscapes while seated indoors documented reduced heart rate and increased parasympathetic activity in participants during that experience [51], making nature viewing VR interventions a promising avenue for relieving stress in health care providers that have limited break-time access to physical immersion in nature.

In this pilot study, we assessed the effectiveness of a Tranquil Cine-VR simulation on subjective stress in frontline workers employed at COVID-19 treatment units in the United States. The participating frontline healthcare workers viewed a three-minute 360-degree cine-VR simulation of a nature scene via a HMD. Prior to the simulation, one-third of participants met the cut-off for high stress. Immediately following the simulation, only 4% of participants met the cutoff for high stress. Overall, we observed a significant reduction in mean stress scores with a large effect size, suggesting that the Tranquil Cine-VR had a meaningful impact on participants' subjective stress. Post-simulation stress scores did not differ by type of frontline healthcare worker, age range, gender, or prior experience with VR. A larger sample of respondents is underway to examine demographic differences in response to the Tranquil Cine-VR. In summary, findings presented here support the use of the Tranquil Cine-VR simulation as a brief, accessible intervention to reduce subjective stress among frontline healthcare workers in COVID-19 treatment units.

Our findings are consistent with prior research examining the use of virtual environments to promote relaxation. A 2021 systematic review by Riches *et al.* [52] concluded that virtual environments with pleasant, natural stimuli increase relaxation and reduce stress in the short-

term. Ten of the 19 studies included in this review used virtual environments depicting scenes from nature (e.g., lakes, forests, clouds, beaches). For example, Wang *et al.* in 2019 [53] examined the effect of seven different forest environments on psychological and physiological stress levels via second-generation VR glasses of the illusion mirror type. All seven forest environments reduced stress levels, although each image produced slightly different effects on stress reduction, suggesting that what is included in an image may influence psychological improvement [53]. Key differences between Wang *et al.* [53] and our study include our use of VR HMDs to achieve full immersion and our 360-degree video capture of the nature scene with accompanying nature sounds.

A study by Valtchanov *et al.* in 2010 [54] compared a computer-generated nature setting to abstract paintings to assess changes in affect, skin-conductance response, and heart rate. In this study, participants completed a stress-induction task prior to viewing the nature setting in a HMD or a slideshow of abstract paintings on a screen in an empty room [54]. Findings suggest that exposure to the computer-generated nature setting decreased the skin-conductance response and increased positive affect [54]. In our study, we used 360-degree video capture of a real nature scene, but we did not include a control condition or measure physiological responses to the VR, as in the present context our primary goal was to relieve provider fatigue with a minimum of contact or additional burden to the healthcare workers in our study.

Lastly, a study by Annerstedt *et al.* in 2013 [55] explored subjective and objective stress and recovery responses following a stress test via three conditions: 1) a virtual forest with nature sounds; 2) the same virtual forest with no sounds; and 3) a control condition with no virtual forest or sounds. The virtual forest with nature sounds induced parasympathetic activity and more efficient stress recovery compared to the virtual forest with no sounds and the control condition [55]. Of note, the Annerstedt *et al.* [55] study was conducted using a CAVE™ system with three rear-projected walls (4 m × 3 m) and a floor projection to show the three virtual conditions (EON Development Inc.) and not HMDs. Similar to Annerstedt *et al.* [55] study, we included a virtual nature scene with accompanying sounds; however, for the reasons noted above we did not measure physiological responses or include comparison conditions. Future VR should integrate the strengths from all of these studies to create multiple 360-degree videos of real nature scenes with congruent sounds to be viewed on HMDs. Further comparisons of Tranquil Cine- VR or nature-based VR to control conditions will permit assessment of effectiveness via psychological and physiological measures.

## Limitations

Study limitations included sample size and gender participant self-selection, self-reported data, and the brief VAS assessment. More than two-thirds of the participants self-identified as women. Although gender distribution in our study was unbalanced, this likely reflects the gender distribution at the three COVID-19 units. According to a recent analysis of the 2019 American Community Survey data from the U.S. Census Bureau, 77% of healthcare workers are women [56]. We did not request OhioHealth employment records to confirm the gender distribution in these COVID-19 units as personnel shifted across units to accommodate the surge in COVID-19 cases. This further limits our ability to determine the total population size, and in turn, the response rate for the study. The small sample size also decreased the power in our sample, which in turn, increased the probability of a Type II error. Specifically, the small number of participants limited our ability to detect potential differences by provider type, gender, age range, and prior VR experience. Research with a larger sample size is underway to identify demographic differences in response to the Tranquil Cine-VR experience. Our findings may also be susceptible to selection bias, as healthcare workers who volunteered to

participate may have been more stressed and/or receptive to a VR intervention. In addition, the responses may be susceptible to selection bias if participants felt undue pressure to provide positive feedback on the Tranquil Cine-VR simulation, although participant education materials emphasized the importance of accurate self-reporting. Finally, this study presents findings from the VAS pre-post assessment immediately following the Tranquil Cine-VR simulation. We did not include additional measures of stress and burnout, a long-term follow-up, or a control group for comparison. Future research should employ a randomized control trial to assess the impact of the Tranquil Cine-VR compared to a control condition with both subjective and objective measures of stress and burnout collected at multiple time points; these were deemed beyond the scope of this study that aimed to provide in-the-moment stress relief to highly fatigued frontline healthcare workers during a pandemic surge.

## Conclusion

Psychological support is a critical need for frontline healthcare workers during the COVID-19 pandemic. Our findings support the use of the Tranquil Cine-VR to reduce subjective stress in the short-term among frontline healthcare workers. The Tranquil Cine-VR is a brief, accessible intervention that frontline healthcare workers can use during their shift work. Long-term impacts of Tranquil Cine-VR on subjective stress and other psychosocial constructs (e.g., anxiety, depressive symptoms, burnout) are promising areas for future research.

## Author Contributions

**Conceptualization:** Elizabeth Beverly, Matthew Love, Carrie Love, Michelle Pershing, Nancy Stevens.

**Data curation:** Elizabeth Beverly, Matthew Love, Carrie Love, Michelle Pershing, Nancy Stevens.

**Formal analysis:** Elizabeth Beverly.

**Investigation:** Elizabeth Beverly, Laurie Hommema, Kara Coates, Gary Duncan, Brad Gable, Thomas Gutman, Matthew Love, Carrie Love, Michelle Pershing, Nancy Stevens.

**Methodology:** Elizabeth Beverly, Laurie Hommema, Kara Coates, Thomas Gutman, Matthew Love, Carrie Love, Michelle Pershing, Nancy Stevens.

**Project administration:** Elizabeth Beverly, Laurie Hommema, Kara Coates, Gary Duncan, Brad Gable, Thomas Gutman, Matthew Love, Michelle Pershing, Nancy Stevens.

**Resources:** Laurie Hommema, Thomas Gutman, Matthew Love, Carrie Love, Michelle Pershing, Nancy Stevens.

**Software:** Elizabeth Beverly, Matthew Love, Carrie Love, Michelle Pershing.

**Supervision:** Laurie Hommema, Kara Coates, Gary Duncan, Brad Gable, Thomas Gutman.

**Validation:** Elizabeth Beverly, Matthew Love, Nancy Stevens.

**Visualization:** Laurie Hommema, Kara Coates, Gary Duncan, Brad Gable, Thomas Gutman, Matthew Love, Nancy Stevens.

**Writing – original draft:** Elizabeth Beverly, Laurie Hommema, Kara Coates, Gary Duncan, Brad Gable, Thomas Gutman, Matthew Love, Carrie Love, Michelle Pershing, Nancy Stevens.

**Writing – review & editing:** Elizabeth Beverly, Laurie Hommema, Kara Coates, Gary Duncan, Brad Gable, Thomas Gutman, Matthew Love, Carrie Love, Michelle Pershing, Nancy Stevens.

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
