## [Decision Letter · Decision Letter 0]

10 Dec 2021

PONE-D-21-32206A Tranquil Virtual Reality Experience to Reduce Subjective Stress among COVID-19 Frontline Healthcare WorkersPLOS ONE

Dear Authors,

Thank you for submitting your manuscript to PLOS ONE. After careful consideration, we feel that it has merit but does not fully meet PLOS ONE’s publication criteria as it currently stands. Therefore, we invite you to submit a revised version of the manuscript that addresses the points raised during the review process.

We look forward to receiving your revised manuscript.

Kind regards,

Kelvin Ian Afrashtehfar, M.Sc., D.D.S.,Dr. med. dent., FRCDC

Academic Editor

PLOS ONE

Journal Requirements:

a) Did participants provide their written or verbal informed consent to participate in this study? 

b) If consent was verbal, please explain i) why written consent was not obtained, ii) how you documented participant consent, and iii) whether the ethics committees/IRB approved this consent procedure

Additional Editor Comments (if provided):

Dear Authors,

Kindly follow the reviewer' suggestains carefully.

Thank you,

The Academic Editor

Reviewers' comments:

Reviewer's Responses to Questions

**Comments to the Author**

1. Is the manuscript technically sound, and do the data support the conclusions?

Reviewer #1: Yes

2. Has the statistical analysis been performed appropriately and rigorously? 

Reviewer #1: No

3. Have the authors made all data underlying the findings in their manuscript fully available?

Reviewer #1: Yes

4. Is the manuscript presented in an intelligible fashion and written in standard English?

Reviewer #1: Yes

5. Review Comments to the Author

Reviewer #1: The authors' attempt to reduce the COVID-19 frontline healthcare workers using a notable and interesting technique is of significant interest to the current scenario. This type of study should be encouraged so that it might help in fighting the current pandemic. The current study is well designed and written. However, I have a major concern about the sample size. The sample size is extremely small. For example, the authors found the statistical significance in post-simulation stress to the participants aged 55-64 years compared to participants in the 18-24 years, 35-44 years, and 45-54 years age group. However, the number of participants in 55-64 years is only 6. Therefore, I am really concerned about this statistical significance. Again, the person in the 25-34 years group usually has small children. The authors should clearly address this issue in the discussion section.

6. PLOS authors have the option to publish the peer review history of their article (what does this mean?). If published, this will include your full peer review and any attached files.

Reviewer #1: No

---

## [Author Response · Author response to Decision Letter 0]

21 Dec 2021

Journal Requirements:

Response: Thank you. We have revised the manuscript to follow the recommended style requirements (i.e., headings, figure citations, table citations, and file naming). 

a) Did participants provide their written or verbal informed consent to participate in this study? b) If consent was verbal, please explain c) why written consent was not obtained, d) how you documented participant consent, and e) whether the ethics committees/IRB approved this consent procedure?

Response: We amended our ethics statement in the “Data collection” section of our Materials and methods section. All participants provided verbal informed consent prior to participation in the assessment. We selected verbal consent because the study was conducted in three different COVID-19 units, and we wanted to limit the use of materials to prevent the spread of the virus. Further, we did not have access to locked filing cabinets on the COVID-19 units to store written informed consent forms per our institutions’ IRB protocols. Participants either read the verbal informed consent form on an iPad or had the verbal informed consent form read to them by a well-being partner. Workers who provided their verbal consent to the well-being partner were provided an iPad to complete the anonymous, electronic REDCap survey. The verbal consent was documented by the well-being partner. Both the OhioHealth Office of Human Subjects Protections (Institutional Review Board #1734245-1) and the Ohio University Office of Research Compliance (Institutional Review Board #21-D-28) approved the verbal consent process.

Reviewers' comments:

Reviewer's Responses to Questions

Comments to the Author

1. Is the manuscript technically sound, and do the data support the conclusions?

Reviewer #1: Yes

Response: We thank the reviewer for the positive comments.

2. Has the statistical analysis been performed appropriately and rigorously? 

Reviewer #1: No

Response: We agree with the reviewer that our sample size is very small. We reran our statistical analyses using nonparametric tests to account for non-normal distribution of the demographic variables. In addition, we added content to the Limitations section addressing the small sample size and the increased risk of a Type II error.

3. Have the authors made all data underlying the findings in their manuscript fully available?

Reviewer #1: Yes

4. Is the manuscript presented in an intelligible fashion and written in standard English?

Reviewer #1: Yes

5. Review Comments to the Author

Reviewer #1: The authors' attempt to reduce the COVID-19 frontline healthcare workers using a notable and interesting technique is of significant interest to the current scenario. This type of study should be encouraged so that it might help in fighting the current pandemic. The current study is well designed and written. However, I have a major concern about the sample size. The sample size is extremely small. For example, the authors found the statistical significance in post-simulation stress to the participants aged 55-64 years compared to participants in the 18-24 years, 35-44 years, and 45-54 years age group. However, the number of participants in 55-64 years is only 6. Therefore, I am really concerned about this statistical significance. Again, the person in the 25-34 years group usually has small children. The authors should clearly address this issue in the discussion section.

Response: We thank the reviewer for the positive comments about the study design and writing quality of the manuscript. We agree that the small sample size is a limitation to our study. We have included additional information addressing the small sample size in the Limitations section. Specifically, we address the limitations of a small sample size in our sample with decreased power and the increased risk of a Type II error in the detection of differences by the demographic variables. In addition, we reran our statistical analyses using non-parametric tests to account for the non-normal distribution. We conducted Kruskal-Wallis tests in replace of the one-way ANOVA analyses. The non-parametric tests revealed no statistical difference with any demographic variables. Also, we removed the Chi-square tests examining the high stress cutoffs by demographic variables. We revised our findings in the text. 

6. PLOS authors have the option to publish the peer review history of their article (what does this mean?). If published, this will include your full peer review and any attached files.

Do you want your identity to be public for this peer review? For information about this choice, including consent withdrawal, please see our Privacy Policy.

Reviewer #1: No

---

## [Editor Report · Decision Letter 1]

3 Jan 2022

A Tranquil Virtual Reality Experience to Reduce Subjective Stress among COVID-19 Frontline Healthcare Workers

PONE-D-21-32206R1

Dear Authors,

We’re pleased to inform you that your manuscript has been judged scientifically suitable for publication and will be formally accepted for publication once it meets all outstanding technical requirements.

Kind regards,

Kelvin Ian Afrashtehfar, M.Sc., D.D.S.,Dr. med. dent., FRCDC

Academic Editor

PLOS ONE